# Activation of Liver X Receptors and Peroxisome Proliferator-Activated Receptors by Lipid Extracts of Brown Seaweeds: A Potential Application in Alzheimer’s Disease?

**DOI:** 10.3390/nu15133004

**Published:** 2023-06-30

**Authors:** Nikita Martens, Na Zhan, Gardi Voortman, Frank P. J. Leijten, Connor van Rheenen, Suzanne van Leerdam, Xicheng Geng, Michiel Huybrechts, Hongbing Liu, Johan W. Jonker, Folkert Kuipers, Dieter Lütjohann, Tim Vanmierlo, Monique T. Mulder

**Affiliations:** 1Department of Internal Medicine, Section Pharmacology and Vascular Medicine, Erasmus University Medical Center, 3015 CN Rotterdam, The Netherlands; n.martens@erasmusmc.nl (N.M.); n.zhan@erasmusmc.nl (N.Z.); g.voortman@erasmusmc.nl (G.V.); f.leijten@erasmusmc.nl (F.P.J.L.); connorvanrheenen@gmail.com (C.v.R.); suzannevanleerdam@hotmail.com (S.v.L.); tim.vanmierlo@uhasselt.be (T.V.); 2Department of Neuroscience, Biomedical Research Institute, European Graduate School of Neuroscience, Hasselt University, B-3590 Hasselt, Belgium; 3Key Laboratory of Marine Drugs, Chinese Ministry of Education, School of Medicine and Pharmacy, Ocean University of China, Qingdao 266003, China; gengxicheng@stu.ouc.edu.cn (X.G.); liuhongb@ouc.edu.cn (H.L.); 4Department of Environmental Biology, Center for Environmental Sciences, Hasselt University, B-3590 Diepenbeek, Belgium; michiel.huybrechts@uhasselt.be; 5Department of Pediatrics, Section of Molecular Metabolism and Nutrition, University of Groningen, University Medical Center Groningen, 9713 GZ Groningen, The Netherlands; j.w.jonker@umcg.nl (J.W.J.); f.kuipers@umcg.nl (F.K.); 6European Research Institute for the Biology of Ageing (ERIBA), University of Groningen, University Medical Center Groningen, 9713 AV Groningen, The Netherlands; 7Institute of Clinical Chemistry and Clinical Pharmacology, University Hospital Bonn, D-53127 Bonn, Germany; dieter.luetjohann@ukbonn.de; 8Department of Psychiatry and Neuropsychology, School for Mental Health and Neurosciences, Division Translational Neuroscience, Maastricht University, 6200 MD Maastricht, The Netherlands

**Keywords:** nuclear receptor superfamily, liver X receptors, peroxisome proliferator-activated receptors, lipid metabolism, phytosterols, seaweed, Alzheimer’s Disease

## Abstract

The nuclear liver X receptors (LXRα/β) and peroxisome proliferator-activated receptors (PPARα/γ) are involved in the regulation of multiple biological processes, including lipid metabolism and inflammation. The activation of these receptors has been found to have neuroprotective effects, making them interesting therapeutic targets for neurodegenerative disorders such as Alzheimer’s Disease (AD). The Asian brown seaweed *Sargassum fusiforme* contains both LXR-activating (oxy)phytosterols and PPAR-activating fatty acids. We have previously shown that dietary supplementation with lipid extracts of *Sargassum fusiforme* prevents disease progression in a mouse model of AD, without inducing adverse effects associated with synthetic pan-LXR agonists. We now determined the LXRα/β- and PPARα/γ-activating capacity of lipid extracts of six European brown seaweed species (*Alaria esculenta*, *Ascophyllum nodosum*, *Fucus vesiculosus*, *Himanthalia elongata*, *Saccharina latissima*, and *Sargassum muticum*) and the Asian seaweed Sargassum fusiforme using a dual luciferase reporter assay. We analyzed the sterol and fatty acid profiles of the extracts by GC-MS and UPLC MS/MS, respectively, and determined their effects on the expression of LXR and PPAR target genes in several cell lines using quantitative PCR. All extracts were found to activate LXRs, with the *Himanthalia elongata* extract showing the most pronounced efficacy, comparable to *Sargassum fusiforme*, for LXR activation and transcriptional regulation of LXR-target genes. Extracts of Alaria esculenta, Fucus vesiculosus, and Saccharina latissima showed the highest capacity to activate PPARα, while extracts of *Alaria esculenta*, *Ascophyllum nodosum*, *Fucus vesiculosus*, and *Sargassum muticum* showed the highest capacity to activate PPARγ, comparable to *Sargassum fusiforme* extract. In CCF-STTG1 astrocytoma cells, all extracts induced expression of cholesterol efflux genes (*ABCG1*, *ABCA1*, and *APOE*) and suppressed expression of cholesterol and fatty acid synthesis genes (*DHCR7*, *DHCR24*, *HMGCR* and *SREBF2*, and *SREBF1*, *ACACA*, *SCD1* and *FASN*, respectively). Our data show that lipophilic fractions of European brown seaweeds activate LXRs and PPARs and thereby modulate lipid metabolism. These results support the potential of brown seaweeds in the prevention and/or treatment of neurodegenerative diseases and possibly cardiometabolic and inflammatory diseases via concurrent activation of LXRs and PPARs.

## 1. Introduction

Liver X receptors (LXRs) α (NR1H3) and β (NR1H2) and peroxisome proliferator-activated receptors (PPARs) α (NR1C1) and γ (NR1C3) are members of the nuclear receptor superfamily of ligand-activated transcription factors and are implicated in transcriptional control of a wide range of biological processes [1]. In response to binding their specific ligands, LXRs and PPARs form heterodimers with retinoid X receptors (RXRs) and bind to the LXR or PPAR response elements (LXREs/PPREs) in the promoter region of target genes. LXRs and PPARs are key players in the control of lipid and glucose metabolism and inflammation [2,3]. Disturbances in these biological processes contribute to metabolic, inflammatory, and neurodegenerative disorders, including Alzheimer’s Disease (AD). AD is the most common form of dementia. It is a progressive neurodegenerative disorder manifested by cognitive loss and defined by the presence of amyloid-β (Aβ)-containing plaques, tau-containing neurofibrillary tangles, synaptic degeneration, and neuroinflammation [4,5,6]. LXRs and PPARs are abundantly expressed in metabolically active tissues, including the brain, and their activation has been found to exert neuroprotective effects, making these receptors interesting targets in the treatment of neurodegenerative disorders, such as AD [1,7,8,9].

LXRs (α and β) are oxysterol sensors implicated in the regulation of cholesterol and lipid homeostasis, including lipogenesis and reverse cholesterol transport, and in the regulation of glucose homeostasis and inflammatory processes. Activation of LXRs is thought to protect against AD pathologies by promoting cholesterol turnover in the brain, reducing inflammation, and possibly through its anti-amyloidogenic effects [8,10,11,12,13,14,15,16]. We and others have shown that synthetic LXRα/β agonists, such as T0901317, protect against cognitive decline in animal models of AD, without affecting the amyloid pathology [8,13,14,15,16,17,18,19]. However, the induction of lipogenesis by T0901317 and most other synthetic LXR-agonists also results in adverse effects such as hepatic steatosis and hypertriglyceridemia which has limited their clinical application [20,21,22]. Such adverse effects are not induced by endogenous (oxy)sterols or (oxy)phytosterols [23]. Lipid extracts of the seaweed *Sargassum fusiforme* (*S. fusiforme*), which contain LXR-activating (oxy)phytosterols such as saringosterol, have been found to activate LXRs and prevent cognitive decline in AD mice [24]. Recently, we also identified (3β,22E)-3-hydroxycholesta-5,22-dien-24-one and fucosterol-24,28 epoxide as LXR agonists in *S. fusiforme* [25]. Unlike synthetic LXR agonists, these *S. fusiforme* extracts did not have adverse effects on hepatic or serum lipid levels, making them a promising alternative for clinical use [24,26]. Recently, we found that purified 24(S)-saringosterol can also prevent cognitive decline in AD mice [26], providing further evidence of the potential health benefits of saringosterol-rich macroalgae.

PPARs (α, δ/β, and γ) are also lipid sensors involved in multiple biological processes, including lipid and glucose metabolism, inflammatory processes, and cell differentiation and migration. We here focus on PPARα and PPARγ. Both PPARα and PPARγ are activated by endogenous and dietary fatty acids (FAs) and their derivatives, and by numerous other synthetic or natural ligands [27,28,29,30]. PPARα is the target of the fibrate class of lipid-lowering drugs used as a therapeutic strategy against dyslipidemia [3,31,32], PPARγ is a target of antidiabetic thiazolidinediones [3] and anti-inflammatory compounds [33]. Activation of PPARs may protect against AD by repression of pro-inflammatory pathways or by its anti-amyloidogenic, anti-oxidative, and insulin-sensitizing effects [7,34,35,36,37,38,39,40,41,42,43,44]. PPARα activators were demonstrated to diminish memory decline, reduce Aβ aggregation (WY-14643, 4-phenylbutyrate, gemfibrozil, and cinnamic acid), and reduce tau phosphorylation, astrogliosis, microgliosis, and postsynaptic protein loss (WY-14643 and 4-phenylbutyrate) in AD mouse models [39,45,46,47]. The PPARγ agonist rosiglitazone was also found to diminish memory deficits, reduce Aβ levels and p-Tau aggregates, and ameliorate the cytotoxic amoeboid morphology of microglia in AD mice [34,48,49]. Beneficial effects of PPARγ agonists pioglitazone and rosiglitazone on the cognitive functioning of patients with AD, especially in those with co-morbid diabetes, have also been reported [50,51,52,53,54]. Altogether, these data underline the potential of LXR and PPAR agonists as well as their concurrent activation for therapeutic applications in AD and other metabolic and inflammatory disorders.

While a *S. fusiforme* lipid extract and 24(S)-saringosterol both prevented cognitive decline in AD mice, a reduction in amyloid deposition was observed exclusively after *S. fusiforme* extract administration [24,26]. Based on the literature reporting a reduction in Aβ levels via activation of PPARα or PPARγ, components in *S. fusiforme* other than saringosterol may activate PPARα or PPARγ. The application of *S. fusiforme* originating from the East Asian coast in Europe is complicated by legislative reasons and by the required specific growth conditions. Therefore, we aim to identify European brown seaweeds with LXR and PPAR-activating capacities comparable to that of *S. fusiforme*. In this study, we assessed the efficacy of lipid extracts of six European brown seaweed species for activation of LXR and PPAR and their effect on target gene expression in cultured cells.

## 2. Materials and Methods

### 2.1. Seaweed Species

The brown seaweed species *Himanthalia elongata* (*H. elongata*), *Sargassum muticum* (*S. muticum*), *Alaria esculenta* (*A. esculenta*), *Ascophyllum nodosum* (*A. nodosum*), *Fucus vesiculosus* (*F. vesiculosus*), and *Saccharina latissima* (*S. latissima*) were selected based on their European origin and their saringosterol content. *H. elongata* and *A. esculenta* were harvested in Ireland and provided by The Seaweed Company (Schiedam, The Netherlands). *S. muticum*, *A. nodosum*, *F. vesiculosus*, and *S. latissima* were harvested in The Netherlands and provided by Stichting Zeeschelp (Kamperland, The Netherlands). *S. fusiforme* was harvested in Japan and purchased from Terrasana BV (Leimuiden, The Netherlands). After harvest, the seaweeds were washed in seawater and dried by air.

### 2.2. Preparation of Seaweed Extracts

The dried seaweed samples were finely powdered in a mixer and soaked overnight in a 2:1 (*v*/*v*) chloroform/methanol mixture upon exposure to Ultraviolet-C (UVC) light (wavelengths between 200–280 nm) at room temperature. After 10 min of sonification, these mixtures were filtered using Whatman filter paper. The filtrates were evaporated in a vacuum rotary evaporator at 40 °C. The remaining lipid fractions were washed with 100% ethanol, again evaporated in the rotary evaporator, and dissolved in 100% ethanol to obtain the final lipid extracts.

### 2.3. Sterol Analysis

In the crude seaweeds and seaweed extracts, the concentrations of saringosterol and its precursor fucosterol were determined using gas chromatography/mass spectrometry as previously described [55]. In summary, the seaweed samples were first dried using a speed vacuum dryer to relate sterol concentrations to dry weight (DW). Then, the sterols were extracted from the dried tissues by adding a mixture of chloroform-methanol. A volume of 1 mL of the extracts was evaporated to dryness and mixed with 1 mL distilled water. To extract the neutral sterols, 3 mL of cyclohexane was added twice. The combined cyclohexane phases were evaporated under a stream of nitrogen, and the sterols were dissolved in n-decane. The sterols were then converted to trimethylsilyl ethers (TMSis) and incubated at 60 °C for 1 h [56]. Saringosterol and fucosterol levels were then determined using gas chromatography-mass spectrometry (GC-MS).

### 2.4. Lipomics Analysis

A UPLC MS/MS method was used for the lipidomics analysis of the seaweed extracts. Chromatographic experiments were performed on a column of InfinityLab Poroshell 120 EC-C18 (2.1 × 150 mm, 2.7 μm) using a Thermo UltiMate 3000 UPLCTM system (Thermo Fisher Scientific, Waltham, USA). The column temperature was set at 35 °C with an injection volume of 1 μL, and the mobile phase consisted of solvent A (0.1% formic acid and 10 mM ammonium formate in acetonitrile/water = 60:40) and solvent B (0.1% formic acid and 10 mM ammonium formate in isopropanol/acetonitrile = 90:10) at a flow rate of 0.3 mL/min. The gradient elution procedure was as follows: 0–1.5 min, 40% B; 1.5–10.5 min, 40–85% B; 10.5–14 min, 85% B; 14–14.1 min, 85–100% B; 14.1–15 min, 100% B; 15–15.2 min, 100–40% B; and 15.2–18 min, 40% B [57]. Mass spectrometry was performed on a Q Exactive™ Focus Orbitrap™ (Thermo Fisher Scientific). The instrument was operated using a full MS/dd-MS2 mode detection, in accordance with our previous study [58]. The UPLC-MS/MS raw data file were imported into Progenesis QI (Waters, Milford, CT, USA) for matching, alignment, and normalization. The relative content in this manuscript refers to the relative content of a molecular species in its sub class.

### 2.5. Arsenic and Cadmium Analysis

Seaweed samples were pulverized with zirconium oxide balls in a jar using a Retsch Vibration mill MM 2000 and divided into three replicates for each species. Next, approximately 100 mg for each sample was placed in open heat-resistant glass tubes (SCHOTT DURAN^®^, Rye Brook, NY, USA) and digested at 110 °C in a heating block using 69% HNO_3_ (ARISTAR^®^, Leicestershire, England for trace analysis) three times and the last time using 37% HCl (ARISTAR^®^ for trace analysis). Lastly, the samples were dissolved in a 2% HCl solution (diluted with Milli-Q H_2_O). The concentrations of arsenic (As) and cadmium (Cd) were quantified via inductively coupled plasma-optical emission spectrometry (ICP-OES 710, Agilent Technologies, Amstelveen, The Netherlands).

### 2.6. Cell Culture

Immortalized human endothelial kidney cells (HEK293; Merck, Amsterdam, The Netherlands), human astrocytoma cells (CCF-STTG1; Merck), human neuroblastoma cells (SH-SY5Y; American Type Culture Collection (ATCC)), and human microglia cells (CHME3; a kind gift from prof. Dr. M. Tardieu, Université Paris-Sud, France) were used for the reporter assays and gene expression studies. All cell lines were cultured in DMEM/F-12 medium (ThermoFisher Scientific, Waltham, MA, USA) supplemented with 10% heat-inactivated fetal calf serum (FCS) (Thermo Fisher Scientific) and 1% 10,000 U penicillin/10,000 μg streptomycin/mL (Thermo Fisher Scientific) at 37 °C and 5% CO_2_.

### 2.7. Cell Transfection

The LXR and PPAR-activating capacity of the seaweed extracts was determined in a cell-based reporter assay previously described by Zwarts et al. [59]. For this purpose, 1.0 × 10^6^ cells were plated in T-25 culture flasks and after 24 h transfected by exposing the cells for 24 h to 1000 ng of pcDNA3.1/V5H6 vector containing clones of the full-length cDNAs for the murine nuclear receptors LXRα, LXRβ, PPARα or PPARγ, 1000 ng of vector encoding RXRα and 4000 ng of vectors encoding LXRE or PPRE using FuGENE^®^ 6 reagent (Promega, Leiden, The Netherlands) according to the manufacturer’s instructions. Control conditions included cells transfected with 2000 ng of the RXRα-containing vector and 4000 ng of the LXRE- or PPRE-containing vector, and cells transfected with 2000 ng of an empty pcDNA3.1/V5-HisA vector (Invitrogen, Carlsbad, CA, USA) and 4000 ng of the LXRE- or PPRE-containing vector. All cells were co-transfected with 1000 ng of Renilla to normalize for variation in transfection efficiency.

### 2.8. LXR and PPAR Reporter Assays

Transfected cells were seeded in a 96-well luminescence plate. After 24 h, the cells were incubated for 24 h in phenol red-free DMEM/F-12 medium (ThermoFisher Scientific) with the seaweed extracts (dosages based on saringosterol content), the LXRα/β agonist T0901317 (1 µM; #293754-55-9; Cayman, Ann Arbor, MI, USA), PPARα agonist WY-14643 (50 µM; #50892-23-4; Merck), PPARγ agonist pioglitazone (10 µM; #111025-46-8; Cayman), or the extract/compound solvents ethanol or DMSO. Cells were lysed in 25 μL lysis buffer and the Firefly and Renilla luminescent signals were measured using the Dual-Luciferase^®^ Reporter assay system (Promega) and a Victor X4 plate reader (PerkinElmer, Groningen, The Netherlands). The relative receptor activity was defined as the ratio of Firefly luminescence to Renilla luminescence. The fold change was defined as the ratio of the relative receptor activity of seaweed- or agonist-exposed cells to the relative receptor activity of ethanol-exposed cells. The experiments, with the stimulation performed in triplicate, were repeated ≥ three times.

### 2.9. Quantitative Real-Time PCR

Cells were incubated for 24 h with the seaweed extracts (dosages based on saringosterol content), RXR agonist bexarotene (1 µM; #153559-49-0, Merck), the LXRα/β agonist T0901317 (1 µM), PPARα agonist WY-14643 (50 µM), PPARγ agonist pioglitazone (10 µM), or the extract or compound solvent ethanol or DMSO. Cells were washed with cold phosphate-buffered saline and RNA was isolated using Trizol (Thermo Fisher Scientific) and reverse transcribed to cDNA using the Maxima H Minus First Strand cDNA Synthesis Kit with dsDNase (Thermo Fisher Scientific), according to the manufacturer’s instructions. Quantitative real-time PCR (qPCR) was conducted in duplicate with 10 ng cDNA on a CFX384 Thermal Cycler (Bio-Rad Laboratories) using the PowerTrack™ SYBR Green Master Mix (Applied Biosystems) and the following cycling conditions: 95 °C for 2 min and 40 cycles of [95 °C for 15 s, 60 °C for 60 s]. The intron-spanning primers for qPCR were designed with Primer-BLAST [60]. Primer sequences are listed in Table 1. Relative quantification of the gene expression was accomplished with the comparative Ct method. The data were normalized to five reference genes (*ACTB*, *B2M*, *HPRT1*, *SDHA,* and *YWHAZ*) and expressed as fold change relative to the EtOH or DMSO control. The experiments were performed three times.

### 2.10. Statistical Analysis

The data are presented as mean ± SD. Extreme values were excluded using Dixon’s principles of exclusion of extreme values [61,62]. Statistical analyses were performed on the data of the LXR and PPAR reporter assays using GraphPad Prism 8. The D’Agostino-Pearson normality test was used to test normal distribution. The fold change values (treatment vs. ethanol control) were analyzed using a Kruskal–Wallis test. Significance levels are denoted as follows: * *p* ≤ 0.05, ** *p* ≤ 0.01, *** *p* ≤ 0.001.

## 3. Results

### 3.1. Characteristics of the Tested Seaweeds and Seaweed Extracts

Because of the observed beneficial effects of *S. fusiforme* in models for AD and atherosclerosis [24,63], we searched for European brown seaweeds with similar effects. Six European brown seaweed species were analyzed for their saringosterol and fucosterol concentrations and compared with *S. fusiforme* (Table 2). Crude *F. vesiculosus*, *S. muticum*, and *H. elongata* contained the highest saringosterol concentrations, comparable to that of *S. fusiforme*. These seaweeds as well as *A. nodosum* also contained the highest fucosterol concentrations. The extract of *H. elongata* was found to contain the highest concentration of saringosterol, comparable to the concentrations found in *S. fusiforme* extract*,* while extracts of *F. vesiculosus*, *S. muticum*, and *A. nodosum* contained the highest fucosterol concentrations. Next, we conducted a targeted analysis of the composition of glycerolipids. The molecular characteristics of phospholipids, glyceroglycolipids, diglycerides, triglycerides, and fatty acids in each seaweed extract are summarized in Appendix A.

Predominant FAs found in the extracts were the saturated FAs myristic acid (C14:0) and palmitic acid (C16:0), monounsaturated variants palmitoleic acid (C16:1) and oleic acid (C18:1), and polyunsaturated FAs (PUFAs) linoleic acid (C18:2/C18:3), parinaric acid (C18:4), arachidonic acid (C20:4) and EPA (C20:5). The odd-chain saturated fatty acid heptadecanoic acid (C17:0) was also found to be one of the predominant molecular species in all the seaweed extracts tested, with the exception of the *A. esculenta* extract. These natural FAs and eicosanoids, and other lipids found in the extracts, can serve as ligands for PPARs (Appendix A).

Because of the known high arsenic concentrations in *S. fusiforme* that limits the amount that can be consumed safely, we determined arsenic concentrations in all seaweeds. As expected, *S. fusiforme* contained the highest concentration of arsenic (67.26 ± 3.34 mg/kg DW), followed by *S. muticum* (28.95 ± 1.14 mg/kg DW), *S. latissima* (24.62 ± 1.59 mg/kg DW), *A. esculenta* (16.79 ± 1.7 mg/kg DW), *H. elongata* (13.58 ± 0.37 mg/kg DW), *F. vesiculosus* (10.37 ± 0.49 mg/kg DW), and *A. nodosum* (7.64 ± 0.32 mg/kg DW). Cadmium was detected in *S. fusiforme* (1.03 ± 0.02 mg/kg DW), but not in the other seaweeds.

### 3.2. LXR Activating Capacity

The dosage of the lipid extracts in the cell experiments was based on the saringosterol concentrations, with the highest dose being the maximal dose tolerated by the cells (dilution factors presented in Appendix A). The lipid extracts of all the seaweeds activated LXRα and LXRβ, although to a different extent and in a cell type-specific manner (Figure 1, Figure 2 and Figure 3 and Appendix A). LXRs were mostly activated by *H. elongata* comparable to *S. fusiforme,* followed by *S. muticum* and *S. latissima*. The extracts of *F. vesiculosus*, *A. esculenta*, and *A. nodosum* showed LXR activation in HEK and CHME3 cells, exclusively.

### 3.3. PPARα and PPARγ Activation by the Seaweed Extracts

PPARα was most strongly activated by extracts of *A. esculenta*, *F. vesiculosus*, and *S. latissima* (Figure 4 and Appendix A), while activation of PPARγ was strongest by the extracts of *A. esculenta, A. nodosum*, *F. vesiculosus*, and *S. muticum,* overall, to a lesser extent than PPARα (Figure 5 and Appendix A). The relative increase in activation of both PPARα and PPARγ was found to be most pronounced in SH-SY5Y cells (Figure 4, Figure 5 and Figure 6).

### 3.4. Effect of the Seaweed Extracts on the Expression of LXR and PPAR-Target Genes

We examined the effects of the seaweed extracts on gene expression involved in cholesterol efflux and lipid synthesis in astrocytoma cells (CCF-STTG1), given the crucial role of astrocytes in the cerebral cholesterol metabolism and their contribution to the supply of cholesterol to neurons. All extracts, although to a different extent, induced the expression of the LXR target genes *ABCG1*, *ABCA1*, and *APOE* involved in cholesterol efflux (Figure 7). The extracts suppressed the expression of *DHCR7*, *DHCR24*, *HMGCR*, and *SREBF2* involved in cholesterol synthesis (Figure 8) as well as *SREBF1*, *ACACA*, *SCD1,* and *FASN* involved in fatty acid synthesis (Figure 9). The expression of *DHCR24* and *HMGCR* was also reduced by bexarotene and T0901317 (Figure 8); however, as expected, both bexarotene and T0901317 increased the expression of the tested fatty acid synthesis genes (Figure 9). The expression of the glial fibrillary acidic protein (*GFAP*), the gene encoding an astrocytic structural protein indicative of astrogliosis, was decreased by the extracts, while T0901317, WY14643, and pioglitazone increased its expression (Figure 10).

## 4. Discussion

LXRs and PPARs are recognized as interesting therapeutic targets for cardiometabolic, inflammatory, and neurodegenerative disorders such as AD because their activation can beneficially impact the pathology of these diseases via modulation of lipid metabolism and/or inflammatory processes. Therefore, we have screened lipid extracts of six European seaweeds for their ability to activate LXRs and PPARs. We demonstrate that all six seaweeds tested contained the LXR-activating oxyphytosterol saringosterol as well as multiple PPAR-activating FAs. The *H. elongata* extract with the highest saringosterol content, but not the highest fucosterol concentration, displayed the highest efficacy for activation of LXRs and subsequently for modulation of LXR-target gene expression, comparable to the *S. fusiforme* extract. *S. muticum*, *S. latissima*, *F. vesiculosus*, *A. esculenta*, and *A. nodosum* also activated LXRs and regulated LXR-target gene expression, albeit to a lesser extent than *H. elongata.* PPARα was most strongly activated by *A. esculenta*, *F. vesiculosus* and *S. latissima*, and PPARγ by *A. esculenta*, *A. nodosum*, *F. vesiculosus* and *S. muticum*. The concurrent activation of LXRs and PPARs by the seaweed extracts may combine the beneficial effects of both. These data are supportive of the potential of brown seaweeds for prevention and/or treatment of neurodegenerative, and possibly also cardiometabolic and inflammatory diseases.

Accumulating evidence suggests that disturbances in brain cholesterol homeostasis are linked to AD pathogenesis. An altered cholesterol turnover rate and altered intra- and intercellular distribution of cholesterol in the brain rather than altered cholesterol levels seem to be involved in neuropathologies [64,65]. LXR activation is believed to prevent AD progression by enhancing cholesterol efflux through the secretion of ApoE-containing lipoprotein-like particles, which provide neurons with cholesterol and other lipids that support synaptic plasticity and neuronal regeneration after injury [66]. These processes are promoted by 24-hydroxycholesterol, which is an endogenous LXR agonist that is formed in neurons from cholesterol via the enzyme CYP46A1. Upregulation of the conversion of cholesterol into 24-hydroxycholesterol via activation of CYP46A1 has been shown to protect against pathologies involved in Alzheimer’s, Huntington’s, and Parkinson’s disease, multiple sclerosis, and amyotrophic lateral sclerosis [67], and it is currently being tested in a phase I trial in AD patients (NCT03706885). A 24-oxidized sterol, 24(S)-Saringosterol, similar to 24(S)-hydroxycholesterol, has the ability to cross the blood–brain barrier, and has previously been shown to prevent cognitive decline in AD mice, possibly by exerting effects similar to 24-hydroxycholesterol [24,26]. Fucosterol can also cross the blood–brain barrier, although to a lesser extent than saringosterol, and can also activate LXRs likely indirectly via upregulation of endogenous LXR agonist desmosterol (unpublished observation). FAs are continuously transported in and out of the brain [68]. Our current data show that LXRs were activated by lipid extracts of brown seaweeds, mostly by the extracts of *S. fusiforme*, *H. elongata*, *S. muticum*, and *S. latissima*. In CCF-STTG1 astrocytoma cells, the extracts increased the expression of genes involved in cholesterol efflux, with the most pronounced effects observed with extracts of *S. fusiforme* and *H. elongata*, which is consistent with the relatively high concentrations of saringosterol in these extracts. By increasing the ABCA1- and ABCG1-promoted secretion of ApoE-containing particles by astrocytes, the extracts may thus promote protective functions related to neuronal regeneration, synaptogenesis, Aβ clearance, and neuroinflammation [69,70,71].

While the extracts induced genes related to cholesterol efflux, they decreased the expression of genes involved in cholesterol synthesis and fatty acid synthesis in CCF-STTG1 cells. The inhibition of SREBP activation is in line with previous findings showing that (oxy)sterols facilitate the binding of SREBP cleavage-activating protein (SCAP) to Insigs preventing SCAP/SREBP binding and the subsequent SREBP activation. In this way, (oxy)sterols reduce lipogenesis [72]. This feedback mechanism can be mediated by endogenous (oxy)sterols such as cholesterol and 25-hydroxycholesterol [73], but possibly also by exogenous (oxy)sterols such as those contained in seaweed. We cannot rule out a contribution of RXR homodimers to the observed effects of the seaweed extracts because the RXR agonist bexarotene also upregulated cholesterol efflux and downregulated cholesterol synthesis genes *DHCR24* and *HMGCR* similar to T0901317 and the extracts.

Although purified 24(S)-saringosterol was able to prevent cognitive decline in AD mice [26], PPARα and PPARγ activation may also have contributed to the previously reported prevention of amyloid pathology and cognitive decline in AD mice [7,34,39,45,46,47,48,49,74]. Natural PPAR agonists may exert fewer side effects than full-blown synthetic PPAR agonists such as thiazolidinediones which have serious adverse effects [30]. PPARα and PPARγ were activated by the extracts of *A. esculenta, A. nodosum* and *F. vesiculosus* and to a lesser extent by *S. fusiforme, H. elongata* and *S. latissima*. PPARα is the most promiscuous isoform of the PPARs and interacts with saturated and unsaturated FAs [75]. The FA binding profile of PPARγ is the most restricted of the three isoforms, interacting most efficiently with PUFAs and only weakly with monounsaturated FAs [75]. We found a variety of saturated and unsaturated FAs in the extracts that are known for their PPAR-activating capacity. However, other unknown natural ligands may also be present in the extracts and activate PPARs. The *A. esculenta* extract was the most effective PPARα and -γ activator and also contained the highest relative amount of total PUFAs, including known PPAR-activating FAs. The cell lines may have responded differently due to variations in the presence of co-factors.

LXR and PPAR activation has been demonstrated to have anti-inflammatory effects via transrepression of inflammatory transcription factors such as NFĸB and AP-1 [76,77,78]. GFAP is a commonly used marker of astrogliosis related to inflammation and is demonstrated to be elevated in plasma of AD patients [79,80,81]. The reduction in *GFAP* expression observed in CCF-STTG1 cells incubated with seaweed extracts suggests that the extracts may decrease astrogliosis and inflammation, associated with AD and other neurological conditions. However, we have not yet further investigated the potential anti-inflammatory effects of the seaweed extracts induced via LXRα/β and PPARα/γ activation.

One limitation of current experiments is that central nervous system (CNS) cell lines were incubated with the entire seaweed extracts, which contain both known and unknown lipids. However, it is known that only certain lipids, such as (oxy)sterols and FAs, can cross the blood–brain barrier and reach the brain. Therefore, further studies are necessary to verify the effects of tested seaweed extracts on lipid metabolism in the CNS in an in vivo setting, where peripheral effects as well as the crosstalk between the brain and periphery can be explored. It should be noted that extracts were added based on saringosterol concentrations, resulting in comparable but not identical dilutions. While our results demonstrate that the extracts can activate the nuclear receptors LXR and PPAR and affect the expression of several target genes, we cannot draw conclusions on their relative efficiencies. Additionally, we did not consider the harvest period of the seaweeds, which may have resulted in seasonal differences affecting our results, and only total arsenic levels have been analyzed, not the toxic inorganic form separately.

## 5. Conclusions

Our findings indicate that the lipophilic fractions of six different European brown seaweed species possess the ability to activate LXRs and PPARs in both peripheral and CNS cell lines and can influence lipid metabolism in astrocytoma cells. These findings provide a basis for further investigation into their application in the prevention and/or treatment of neurodegenerative, metabolic, and inflammatory diseases. Among the European seaweed species studied, *Himanthalia elongata* demonstrated the highest efficacy for LXR activation, comparable to that of *Sargassum fusiforme*, and comparably regulated cholesterol efflux and lipid synthesis pathways in astrocytoma CCF-STTG1 cells. Previous studies have demonstrated that *Sargassum fusiforme* has a positive impact on cognitive performance and AD pathology in an AD mouse model [24], and therefore, *Himanthalia elongata* could be a promising alternative to *Sargassum fusiforme* for the prevention of AD pathology.

## Figures and Tables

**Figure 1 nutrients-15-03004-f001:**
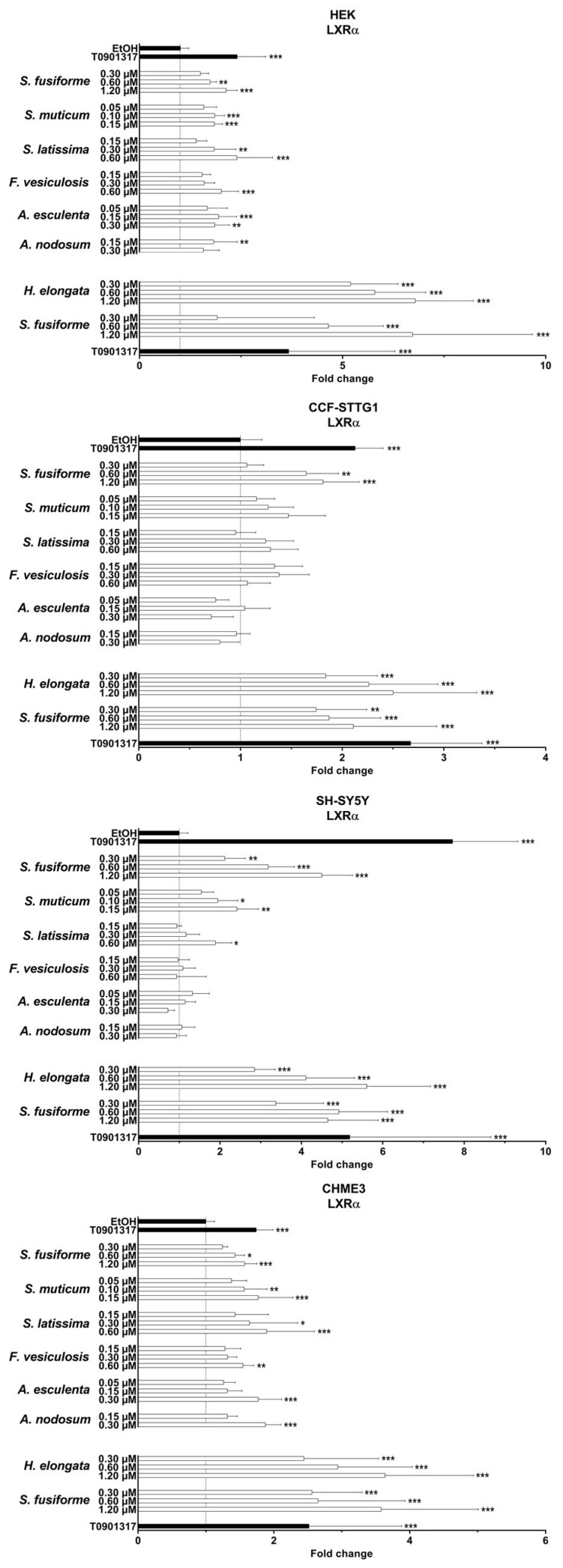
LXRα activation by seaweed extracts. The seaweed extracts were screened for their LXRα activating ability in HEK, CCF-STTG1, SH-SY5Y, and CHME3 cells. Saringosterol concentrations are presented on the *Y*-axis. The fold change values are presented as mean ± SD of three experiments performed in triplicate (n = 9). Significance relative to the ethanol control (Kruskal–Wallis test): * *p* ≤ 0.05, ** *p* ≤ 0.01, *** *p* ≤ 0.001.

**Figure 2 nutrients-15-03004-f002:**
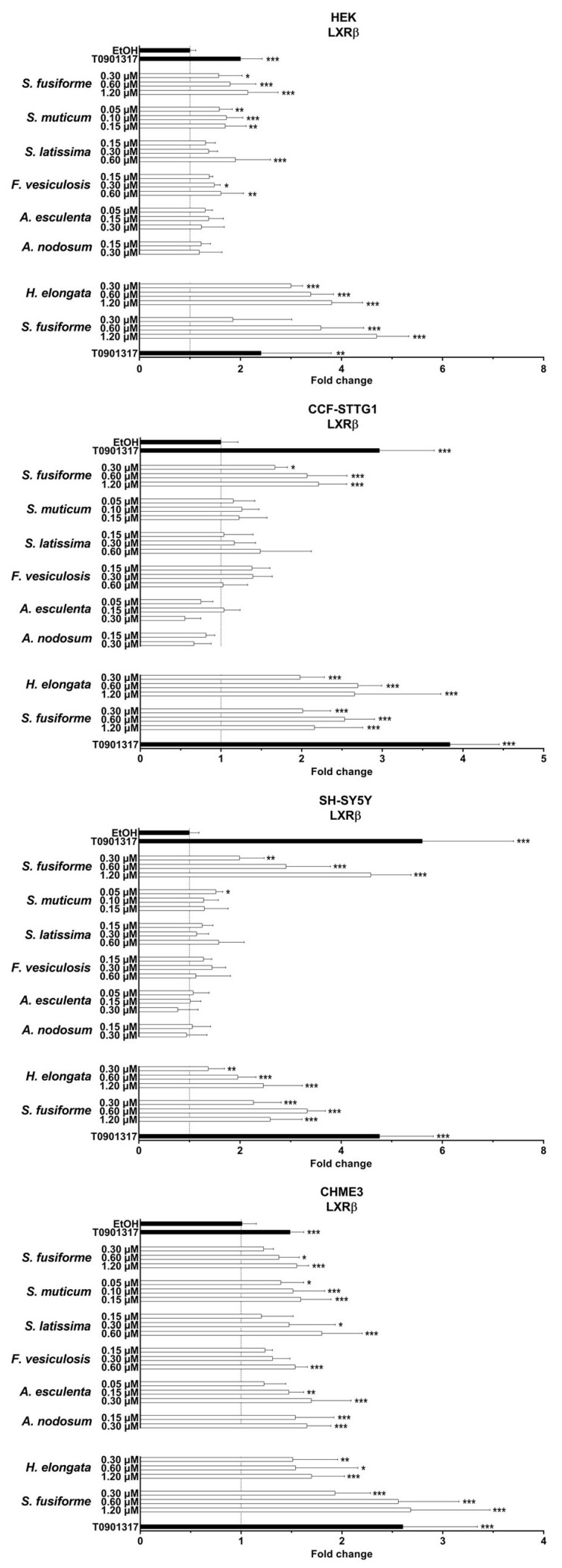
LXRβ activation by seaweed extracts. The seaweed extracts were screened for their LXRβ activating ability in HEK, CCF-STTG1, SH-SY5Y, and CHME3 cells. Saringosterol concentrations are presented on the *Y*-axis. The fold change values are presented as mean ± SD of three experiments performed in triplicate (n = 9). Significance relative to the ethanol control (Kruskal–Wallis test): * *p* ≤ 0.05, ** *p* ≤ 0.01, *** *p* ≤ 0.001.

**Figure 3 nutrients-15-03004-f003:**
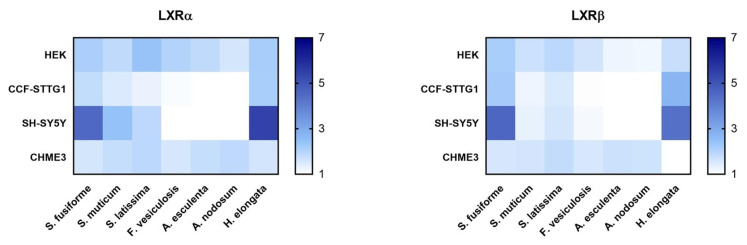
LXRα and LXRβ activation by seaweed extracts. The figures present the fold changes of the highest extract dose of each extract, with the fold change of *H. elongata* corrected for variation in the fold change of *S. fusiforme* in the two experiments.

**Figure 4 nutrients-15-03004-f004:**
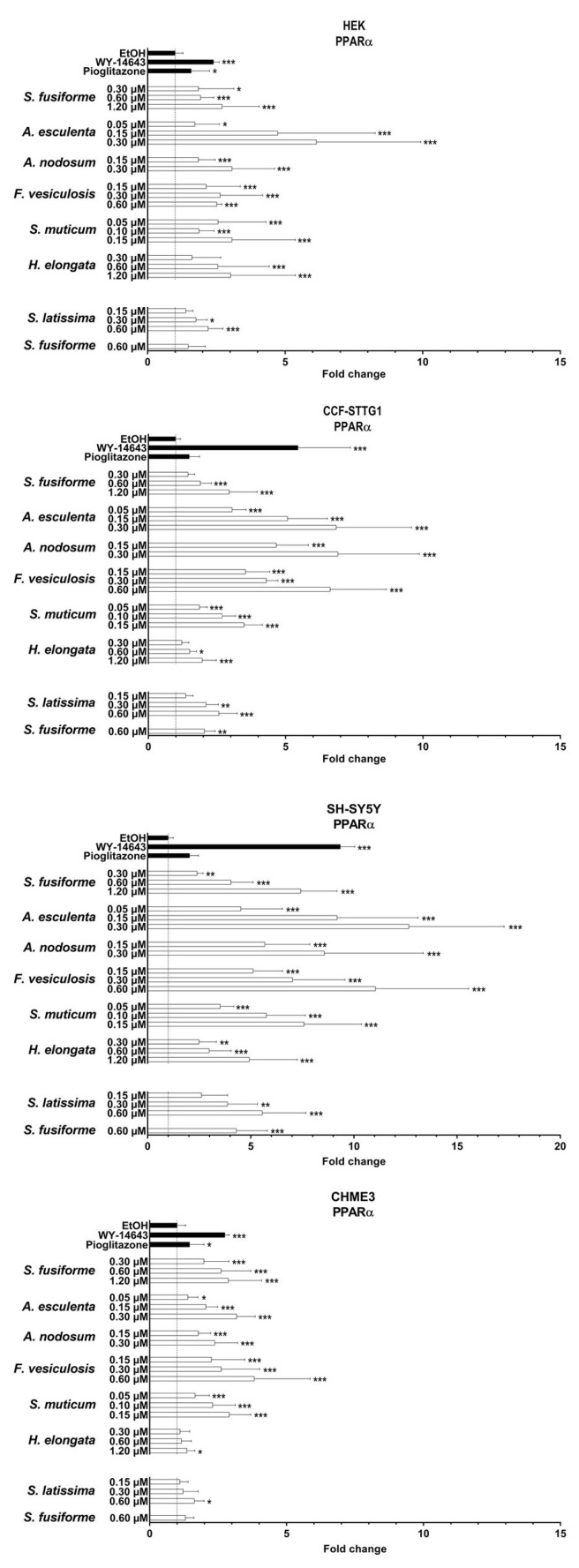
PPARα activation by seaweed extracts. The seaweed extracts were screened for their PPARα activating capacity in HEK, CCF-STTG1, SH-SY5Y, and CHME3 cells. Saringosterol concentrations are presented on the *Y*-axis. The fold change values are presented as mean ± SD of ≥three experiments performed in triplicate (n ≥ 9). Significance relative to the ethanol control (Kruskal–Wallis test): * *p* ≤ 0.05, ** *p* ≤ 0.01, *** *p* ≤ 0.001.

**Figure 5 nutrients-15-03004-f005:**
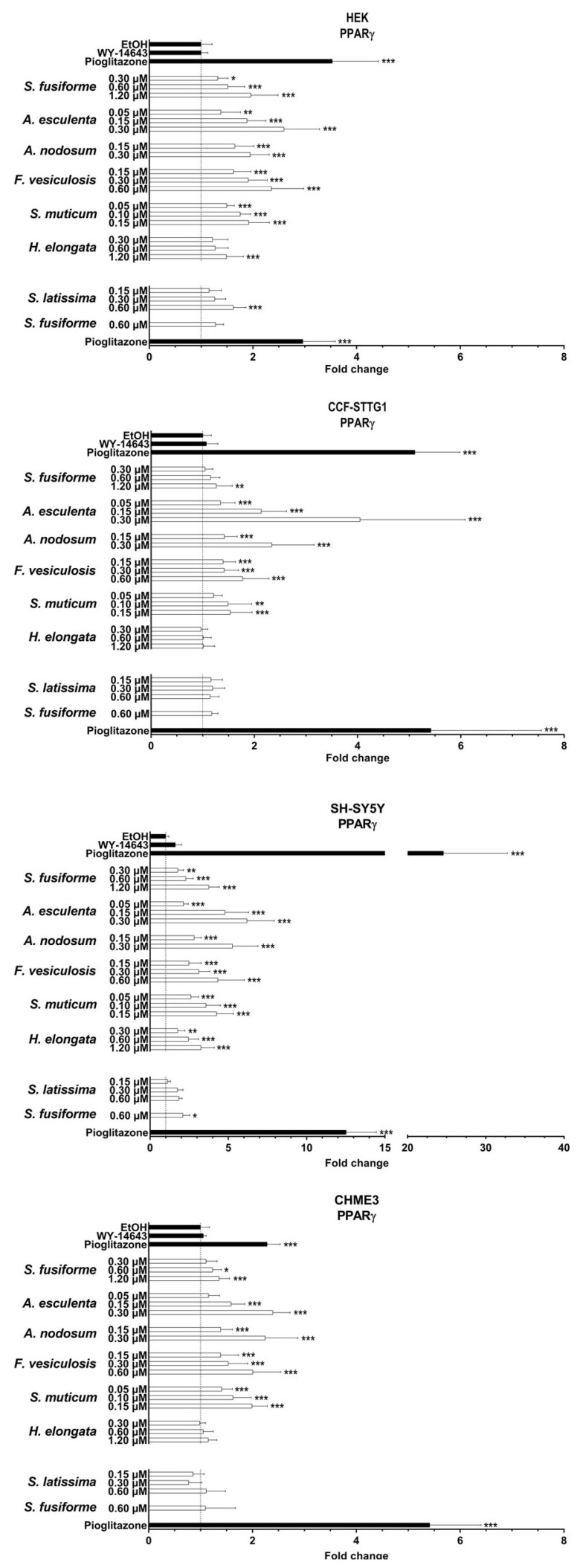
PPARγ activation by seaweed extracts. The seaweed extracts were screened for their PPARγ activating capacity in HEK, CCF-STTG1, SH-SY5Y, and CHME3 cells. Saringosterol concentrations are presented on the *Y*-axis. The fold change values are presented as mean ± SD of ≥three experiments performed in triplicate (n ≥ 9). Significance relative to the ethanol control (Kruskal–Wallis test): * *p* ≤ 0.05, ** *p* ≤ 0.01, *** *p* ≤ 0.001.

**Figure 6 nutrients-15-03004-f006:**
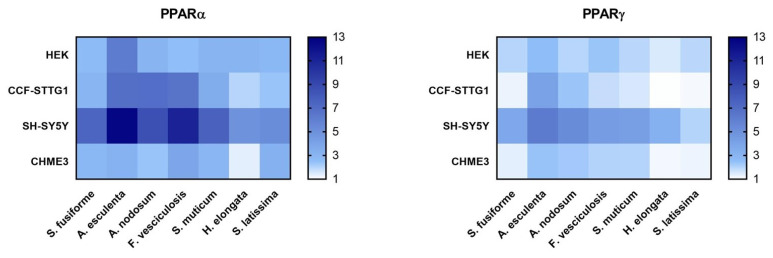
PPARα and PPARγ activation by seaweed extracts. The figures present the fold changes of the highest extract dose of each extract, with the fold change of *S. latissima* corrected for variation in the fold change of *S. fusiforme* in the two experiments.

**Figure 7 nutrients-15-03004-f007:**
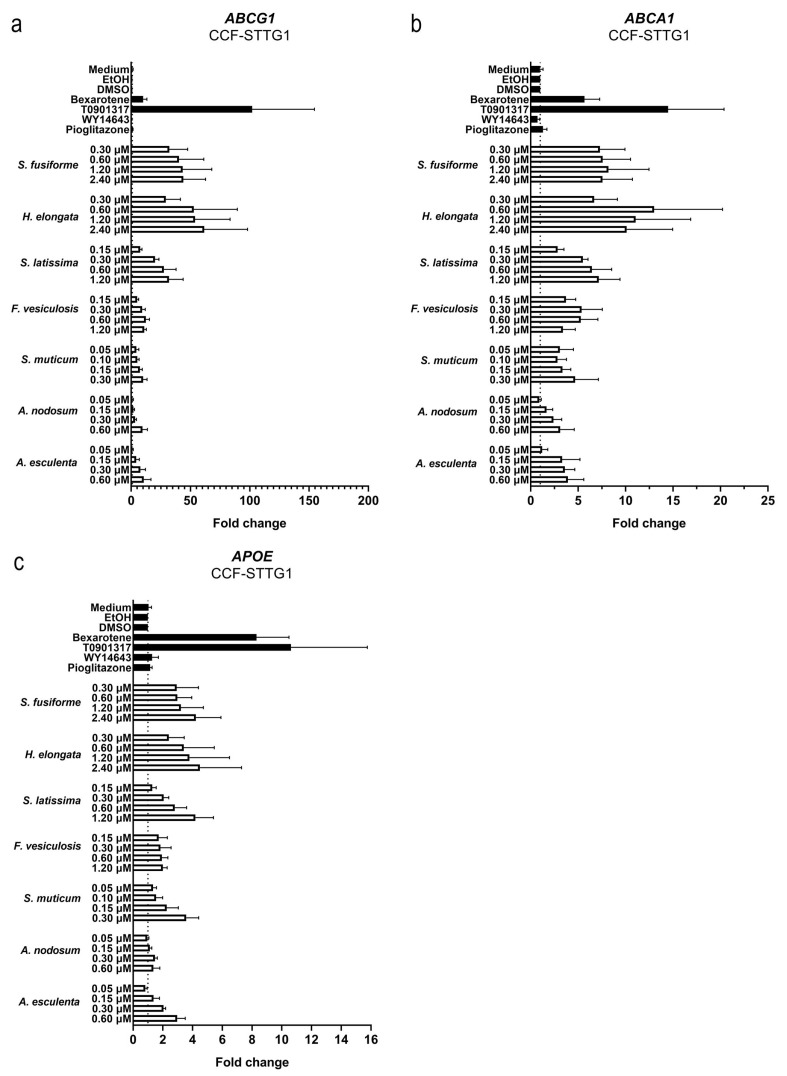
Effect of seaweed extracts on the expression of LXR target genes involved in cholesterol efflux. The expression of cholesterol efflux genes *ABCG1* (**a**), *ABCA1* (**b**), and *APOE* (**c**) in CCF-STTG1 cells was increased by all tested seaweed extracts. Saringosterol concentrations are presented on the *Y*-axis. The experiments were performed three times (n = 3).

**Figure 8 nutrients-15-03004-f008:**
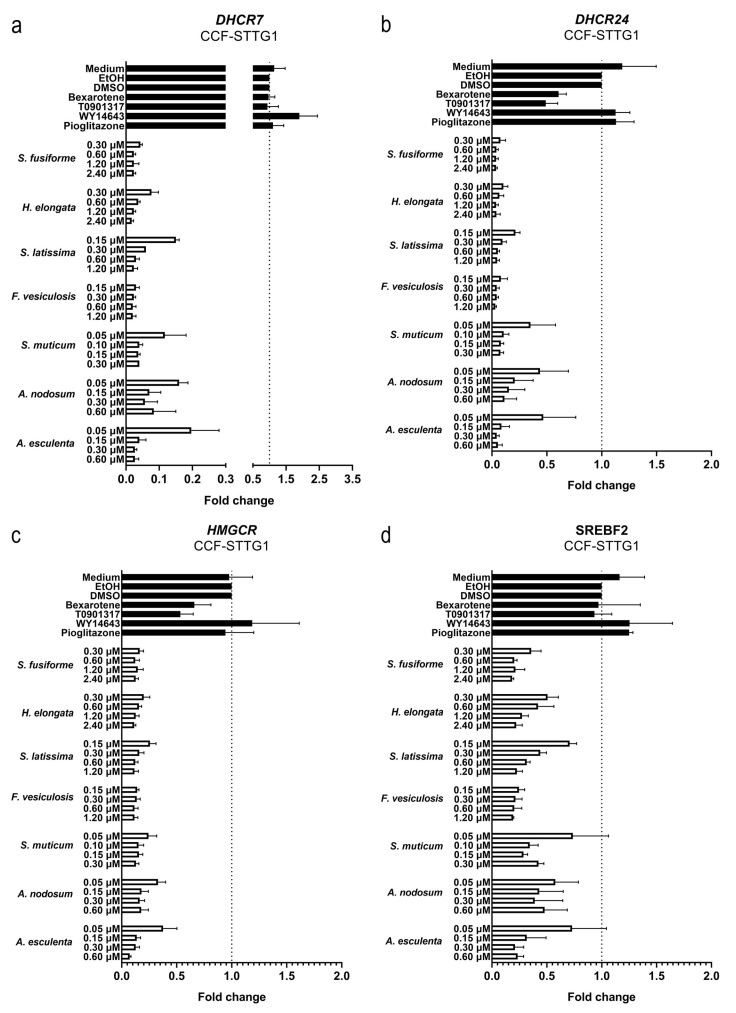
Effect of seaweed extracts on the expression of LXR target genes involved in cholesterol synthesis. The expression of cholesterol synthesis genes *DHCR7* (**a**), *DHCR24* (**b**), *HMGCR* (**c**), and *SREBF2* (**d**) in CCF-STTG1 cells was decreased by all tested seaweed extracts. Saringosterol concentrations are presented on the *Y*-axis. The experiments were performed three times (n = 3).

**Figure 9 nutrients-15-03004-f009:**
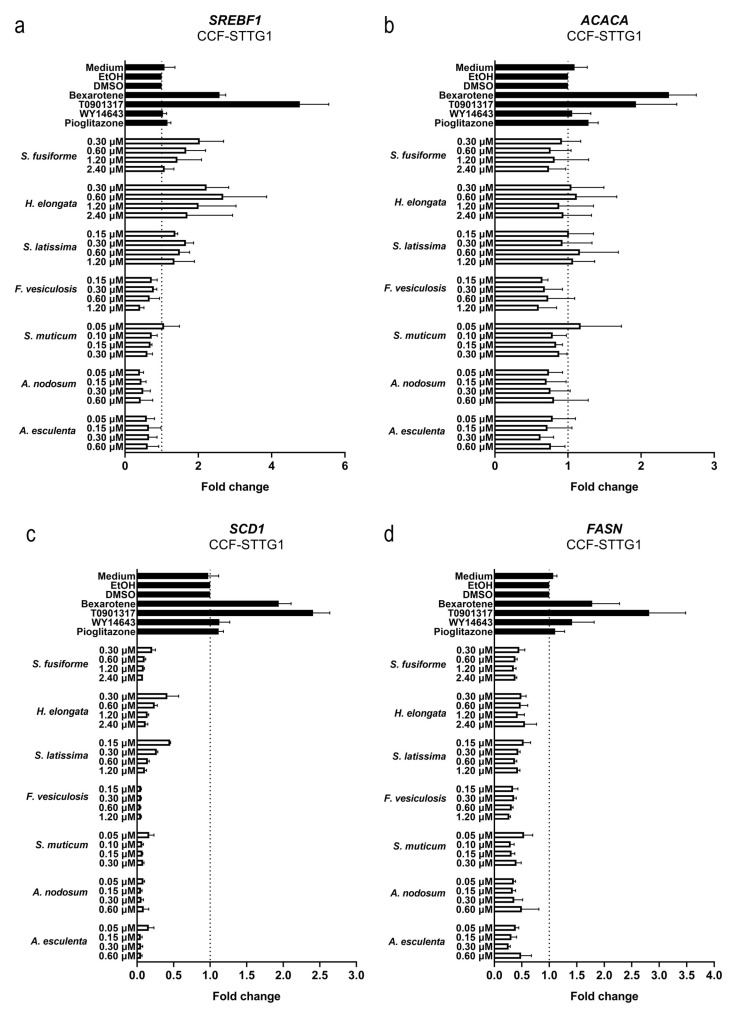
Effect of seaweed extracts on the expression of LXR target genes involved in fatty acid synthesis. The expression of fatty acid synthesis genes *SREBF1* (**a**), *ACACA* (**b**), *SCD1* (**c**), and *FASN* (**d**) in CCF-STTG1 cells was decreased by tested seaweed extracts. Saringosterol concentrations are presented on the *Y*-axis. The experiments were performed three times (n = 3).

**Figure 10 nutrients-15-03004-f010:**
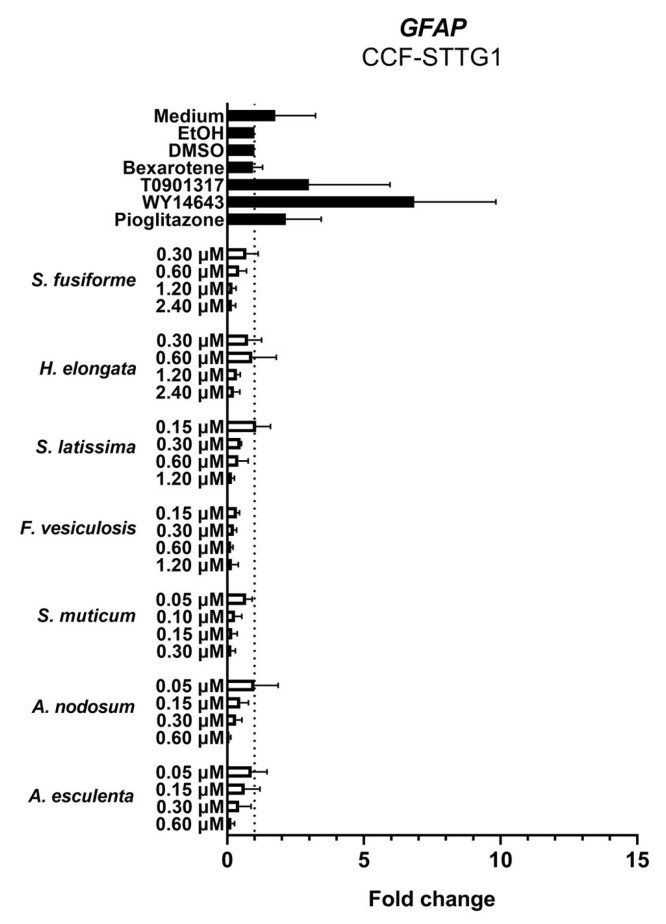
Extracts decreased the expression of *GFAP* in CCF-STTG1 cells. Saringosterol concentrations are presented on the *Y*-axis. The experiments were performed three times (n = 3).

**Table 1 nutrients-15-03004-t001:** Primers and their corresponding forward and reverse nucleotide sequences.

Gene	Gene Name	Primer Sequence
*ABCA1*	ATP Binding Cassette Subfamily A Member 1	F: TCTCTGTTCGGCTGAGCTAC
		R: TGCAGAGGGCATGGCTTTAT
*ABCG1*	ATP Binding Cassette Subfamily G Member 1	F: GGTCGCTCCATCATTTGCAC
		R: GCAGACTTTTCCCCGGTACA
*ACACA*	Acetyl-CoA Carboxylase Alpha	F: GGGTCAAGTCCTTCCTGCTC
		R: GGACTGTCGAGTCACCTTAAGTA
*ACTB*	Actin Beta	F: CTCCCTGGAGAAGAGCTACG
		R: GAAGGAAGGCTGGAAGAGTG
*APOE*	Apolipoprotein E	F: ACCCAGGAACTGAGGGC
		R: CTCCTTGGACAGCCGTG
*B2M*	Beta-2-Microglobulin	F: CTCCGTGGCCTTAGCTGTG
		R: TTTGGAGTACGCTGGATAGCCT
*DHCR7*	7-Dehydrocholesterol Reductase	F: TGGGCCAAGACTCCACCTAT
		R: ACGTGTACAGAAGCACCTGG
*DHCR24*	24-Dehydrocholesterol Reductase	F: GTCTCACTACGTGTCGGGAA
		R: CTCCACACGGACAATCTGTTTC
*FASN*	Fatty Acid Synthase	F: CACAGACGAGAGCACCTTTGA
		R: CAGGTCTATGAGGCCTATCTGG
*GFAP*	Glial Fibrillary Acidic Protein	F: GGCCCGCCACTTGCA
		R: GGGAATGGTGATCCGGTTCT
*HMGCR*	3-Hydroxy-3-Methylglutaryl-CoA Reductase	F: GCAGGACCCCTTTGCTTAGA
		R: GCACCTCCACCAAGACCTAT
*HPRT1*	Hypoxanthine Phosphoribosyltransferase 1	F: TGACACTGGCAAAACAATGCA
		R: GGTCCTTTTCACCAGCAAGCT
*SCD1*	Stearoyl-CoA Desaturase 1	F: GCTGTCAAAGAGAAGGGGAGT
		R: AGCCAGGTTTGTAGTACCTCCT
*SDHA*	Succinate Dehydrogenase Complex Flavoprotein Subunit A	F: TGGGAACAAGAGGGCATCTG
		R: CCACCACTGCATCAAATTCATG
*SREBF1*	Sterol Regulatory Element Binding Transcription Factor 1	F: ACAGCCATGAAGACAGACGG
		R: CAAGATGGTTCCGCCACTCA
*SREBF2*	Sterol Regulatory Element Binding Transcription Factor 2	F: GATCACGCCAACATTCAGCA
		R: GACTTGAGGCTGAAGGACTTGAA
*YWHAZ*	Tyrosine 3-Monooxygenase/Tryptophan 5-Monooxygenase Activation Protein Zeta	F: ACTTTTGGTACATTGTGGCTTCAA
		R: CCGCCAGGACAAACCAGTAT

**Table 2 nutrients-15-03004-t002:** Saringosterol and fucosterol concentrations in crude seaweed and seaweed lipid extracts.

Seaweed Species	Crude Seaweed		Extract	
	Saringosterol (μg/mg DW)	Fucosterol (μg/mg DW)	Saringosterol (mM)	Fucosterol (mM)
*Alaria esculenta*	0.008	0.130	0.2	9.9
*Ascophyllum nodosum*	0.002	0.495	0.2	12.2
*Fucus vesiculosus*	0.034	0.407	0.7	13.0
*Himanthalia elongata*	0.018	0.771	1.8	7.3
*Saccharina latissima*	0.002	0.037	0.7	6.1
*Sargassum fusiforme*	0.026	0.209	1.1	7.0
*Sargassum muticum*	0.032	0.325	0.1	12.4

## Data Availability

Data are available upon request.

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
