# Peer review of "Activation of Liver X Receptors and Peroxisome Proliferator-Activated Receptors by Lipid Extracts of Brown Seaweeds: A Potential Application in Alzheimer’s Disease?"

_nutrients, 2023, doi:10.3390/nu15133004_

Round 1

Reviewer 1 Report

This manuscript shows that the lipophilic fractions of six different European brown seaweed species possess the ability to activate LXRs and PPARs. Accordingly, it can potentially benefit future relevant investigations. Yet, the following additions/clarifications are recommended for the ease of reading and understanding of our future readers.

Based on the data shown in this manuscript, it seems a little premature to cover “the treatment of Alzheimer’s Disease” as indicated in the current article title. To keep such a title, some data obtained from an AD model (in vitro or in vivo, cellular or animal) is strongly recommended. One example could be showing enhancement of the secretion of ApoE-containing lipoprotein-like particles as discussed by the authors. Further, in the last paragraph of the introduction section, the authors indicated that “[w]hile S. fusiforme and 24(S)-Saringosterol both prevented cognitive decline in AD mice, a reduction in amyloid deposition was observed exclusively after S. fusiforme administration (24, 26). Therefore, we hypothesized that other components in S. fusiforme reduce Aβ aggregation, possibly via activation of PPARα or PPARγ.” Yet, I do not see any relevant data in this manuscript testing this hypothesis. Accordingly, this hypothesis is suggested to be relocated to the discussion section to avoid any possible confusion.

Kindly elaborate on why CCF-STTG1 cells were selected for the analysis presented in Section 3.4. Considering CCF-STTG1 represents astrocytoma cells, it would be very interesting to see corresponding results of other brain cell types, such as SH-SY5Y (neurons) and CHME3 (microglia cells). For example, in the discussion section, the authors pointed out that “24(S)-Saringosterol … has previously been shown to prevent cognitive decline in AD mice, possibly by exerting effects similar to 24-hydroxycholesterol,” while “24-hydroxycholesterol, which is an endogenous LXR agonist that is formed in neurons from cholesterol.” Along this line, data from SH-SY5Y representing neurons would particularly interest our future readers.

Further, Figures 1 and 2 are too blurry to read. A version with a higher resolution is required for a detailed review.

The following minor changes would be really appreciated.

(1)    The first sentence in Section 2.8 and the title of Section 3.1 are suggested to be rewritten for clarity.

(2)    The font of the seaweed names in section 2.1 is suggested to be changed to italic for consistency.

(3)    There is an extra “,” in the last sentence of the first paragraph in the discussion section.

Reviewer 2 Report

     Review of - Activation of LXRs and PPARs by lipid extracts of brown seaweeds….2023
This manuscript describes the effects of lipid extracts from various seaweed species on the activation of  (LXRα/β) and peroxisome proliferator-activated receptors (PPARα/γ) to their targeted genes in particular cell lines. The outcome demonstrated the effects on lipid metabolism. Such regulation may have clinical applications. The study is well-described, and the methodological details are presented well.  Writing is very clear and easy to comprehend despite the complexity of the subject material.
Introduction
1.    “Disturbances in these biological processes contribute to metabolic, inflammatory, and neurodegenerative disorders, including Alzheimer’s Disease (AD).”

Maybe here define what type of disturbance such as high or low expression/production?  
The premise of the treatment by seaweed would be that LXRs and PPARs are not being activated in AD and that the seaweed extract activates these receptors. So, in AD are these (LXRs and PPARs) low in activation? Has this been demonstrated?
2.    Examining other seaweed species for arsenic was a good idea, as was noting that levels might be seasonally dependent.
3.    Figures are very hard to read.  Consider breaking each section of the figures into an individual full or half page.
4.    Discussion section - “Accumulating evidence suggests that disturbances in brain cholesterol homeostasis are linked to AD pathogenesis.”

So, is the suggestion that too low of cholesterol is the issue for AD?  It seems as though the next line states that increasing cholesterol is beneficial.

5.    “prevention of AD pathology in Europe.”  Why limit to Europe?

Reviewer 3 Report

Nuclear liver X and peroxisome proliferator-activated receptors are pivotal for regulation cellular lipid metabolism and inflammation. Since their activation has been found to have neuroprotective effect, these receptors became therapeutic targets in case of neurodegenerative diseases. Thus, the work by Martens and collaborators entitled “Activation of LXRs and PPARs by lipid extracts of brown seaweeds: potential application in the treatment of Alzheimer’s Disease?” nicely fits to the current trends and provides the broad scientific community with a bunch of potentially interesting data. The authors determined the capacity of lipid extracts of various brown seaweed species to activate some of the receptors within the two above mentioned families and analyzed sterol and fatty acid profiles of the extracts. In general, the concept of the study is clear and reasonable, the data are carefully documented and the whole story is well described. Therefore, I recommend publishing the work, although only after considering some issues listed below.    

- The authors should pay more attention on the interplay between receptor level and receptor activity. The data showing receptor levels (along the expression of the genes encoding them) should be provided (for example as a supplementary figure). Were the receptor levels (i.e. protein levels, not only transcript levels) the same for all cell lines used? Without this dataset it would be hard to make any comment on putative differences between individual cell lines (see page 9)

- It is highly recommended to show relative content/ratio between major lipid classes (including glycerophospholipids: PA, PG, PE etc.) 

- What was the housekeeping gene(s) employed in Ct method?  

- Some minor issues include: within the last paragraph of the INTRODUCTION the authors should clearly state what form of S.fusiforme (extracted, powdered or prepared in a different way) is the active one; some methodological details are missing (e.g. how the samples were powdered, were the concentrations of sterols expressed per mg of dry weight, details of GC-MS are missing, how the “relative content” was calculated, the definition of “RT” is missing); the text within Figures is barely readable (this is particularly in case of Fig. 1 and 2)  

Round 2

Reviewer 1 Report

Many thanks for the authors’ amendments and patient explanations. Really appreciated them. I only have two minor follow-up concerns.

To smooth the transition from Section 3.3 to Section 3.4 and to emphasize the importance of astrocytes among brain cells, it is suggested to add one or two sentences at the beginning of Section 3.4 elaborating on why CCF-STTG1 cells were selected for the analysis presented in Section 3.4.

Unfortunately, Figures remain very blurry, at least in the manuscript version that I downloaded. Sometimes WORD compresses its embedded images. Accordingly, it may be helpful to upload the original figures in image file type, such as TIFF, one by one to the submission system, if possible.
